# Unifying Activation- and Timing-based Learning Rules for Spiking Neural Networks

**Jinseok Kim**[1]       **Kyungsu Kim**[1]       **Jae-Joon Kim**[1,2]

[1]Department of Creative IT Engineering,
[2]Graduate School of Artificial Intelligence
Pohang University of Science and Technology (POSTECH), Korea
`{jinseok.kim, kyungsu.kim, jaejoon}@postech.ac.kr`

## Abstract

For the gradient computation across the time domain in Spiking Neural Networks (SNNs) training, two different approaches have been independently studied. The first is to compute the gradients with respect to the change in spike activation (activation-based methods), and the second is to compute the gradients with respect to the change in spike timing (timing-based methods). In this work, we present a comparative study of the two methods and propose a new supervised learning method that combines them. The proposed method utilizes each individual spike more effectively by shifting spike timings as in the timing-based methods as well as generating and removing spikes as in the activation-based methods. Experimental results showed that the proposed method achieves higher performance in terms of both accuracy and efficiency than the previous approaches.

## 1   Introduction

Spiking neural networks (SNNs) have been studied not only for their biological plausibility but also for computational efficiency that stems from information processing with binary spikes [1]. One of the unique characteristics of SNNs is that the states of the neurons at different time steps are closely related to each other. This may resemble the temporal dependency in recurrent neural networks (RNNs), but in SNNs direct influences between neurons are only through the binary spikes. Since the true derivative of the binary activation function, or thresholding function, is zero almost everywhere, SNNs have an additional challenge in precise gradient computation unless the binary activation function is replaced by an alternative as in [2].

Due to the difficulty of training SNNs, in some recent studies, parameters trained in non-spiking NNs were employed in SNNs. However, this approach is only feasible by using the similarity between rate-coded SNNs and non-spiking NNs [3, 4] or by abandoning several features of spiking neurons to maximize the similarity between SNNs and non-spiking NNs [5–7]. The unique characteristics of SNNs that enable efficient information processing can only be utilized with dedicated learning methods for SNNs. In this context, several studies have reported promising results with the gradient-based supervised learning methods that takes account of those characteristics [8–13].

Previous works on gradient-based supervised learning for SNNs can be classified into two categories. The methods in the first category work around the non-differentiability of the spiking function with the surrogate derivative [14] and compute the gradients with respect to the spike activation [11–13]. The methods in the second category focus on the timings of existing spikes and computes the gradients with respect to the spike timing [8–10, 15]. Let us call those methods as the activation-based methods and the timing-based methods, respectively. Until now, the two approaches have been thought irrelevant to each other and studied independently.

The problem with previous works is that both approaches have limitations in computing accurate gradients, which become more problematic when the spike density is low. The computational cost of

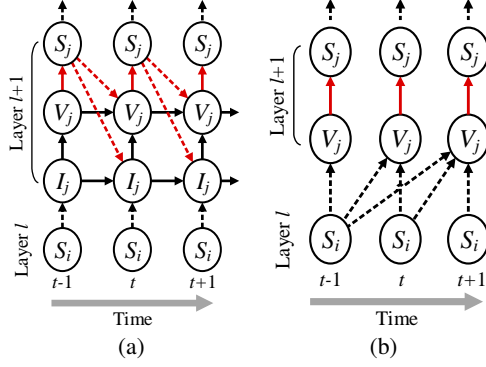

Figure 1: Computational graphs representing (a) the RNN-like description and (b) the SRM-based description of our SNN model. Black solid arrows represent accumulation and decaying. Black dashed arrows represent synaptic integration, red solid arrows represent the spiking function, and red dashed arrows represent reset paths.

the SNN is known to be proportional to the number of spikes, or the firing rates [6, 16, 17]. To make the best use of the computational power of SNNs and use them more efficiently than non-spiking counterparts, it is important to reduce the required number of spikes for inference. If there are only a few spikes in the network, the network becomes more sensitive to the change in the state of each individual spike such as the generation of a new spike, the removal of an existing spike, or the shift of an existing spike. Training SNNs with fewer spikes requires the learning method to be aware of those changes through gradient computation.

In this work, we investigated the relationship between the activation-based methods and the timing-based methods for supervised learning in SNNs. We observed that the two approaches are complementary when considering the change in the state of individual spikes. Then we devised a new learning method called activation- and timing-based learning rule (ANTLR) that enables more precise gradient computation by combining the two methods. In experiments with random spike-train matching task and widely used benchmarks (MNIST and N-MNIST), our method achieved the higher accuracy than that of existing methods when the networks are forced to use fewer spikes in training.

## 2 Backgrounds

### 2.1 Neuron model

We used a discrete-time version of a leaky integrate-and-fire (LIF) neuron with the current-based synapse model. The neuronal states of postsynaptic neuron $j$ are formulated as

$$V_j[t] = \alpha_V(1 - S_j[t-1])V_j[t-1] + \beta_V I_j[t] + \beta_{\text{bias}}V_{\text{bias},j} \tag{1}$$

$$I_j[t] = \alpha_I(1 - S_j[t-1])I_j[t-1] + \beta_I \sum_i w_{i,j}S_i[t] \tag{2}$$

$$S_j[t] = \Theta(V_j[t]) = \begin{cases} 1, & \text{if } V_j[t] \geq \theta \\ 0, & \text{otherwise} \end{cases} \tag{3}$$

where $V_j[t]$ is a membrane potential, $I_j[t]$ is a synaptic current, $S_j[t]$ is a binary spike activation. $w_{i,j}$ is a synaptic weight from presynaptic neuron $i$. $V_{\text{bias},j}$ is a trainable bias parameter. $\Theta$ and $\theta$ are the spiking function and the threshold, respectively. $\alpha_V$ and $\alpha_I$ are the decay coefficients for the potential and the current. $\beta_V$, $\beta_I$, and $\beta_{\text{bias}}$ are the scale coefficients. We call this type of description as the **RNN-like description** since the temporal dependency between variables resembles that in Recurrent Neural Networks (RNNs) [14] (Figure 1a). The term $(1 - S_j[t-1])$ was introduced in $V_j[t]$ and $I_j[t]$ to reset both the potential and the synaptic current. Note that this model can express various types of commonly used neuron models by changing the decay coefficients (Figure A1 in Appendix A).

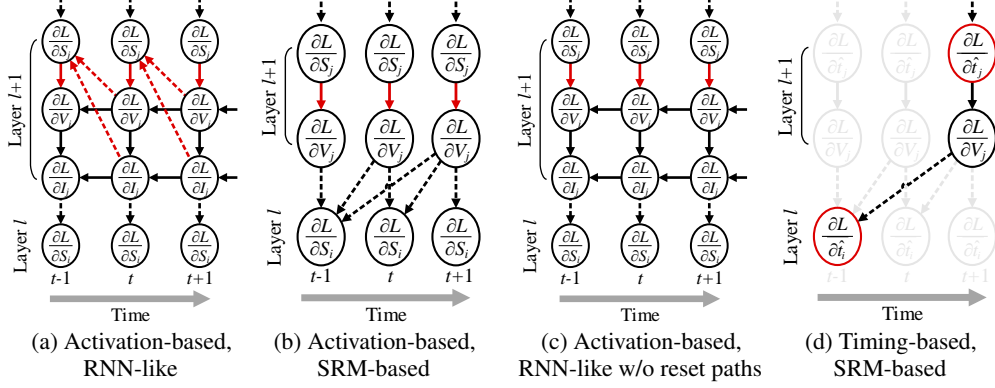

Figure 2: Various types of back-propagation derived from different neuron model descriptions. Black solid arrows, black dashed arrows, red solid arrows, and red dashed arrows represent back-propagation paths for accumulation and decaying, synaptic integration, spiking function, and reset paths, respectively.

The same neuron model can also be formulated using the spike response kernel $\varepsilon[\tau] = \beta_I \beta_V \sum_{k=0}^{\tau} \alpha_I^k \alpha_V^{\tau-k}$ as

$$V_j[t] = \sum_i \sum_{\hat{t}_i \in \mathcal{T}_{i,j,t}} w_{i,j} \varepsilon[t - \hat{t}_i] = \sum_{\tau=\hat{t}_j^{\text{last}}[t]+1}^{t} \sum_i w_{i,j} \varepsilon[t - \tau] S_i[\tau] \tag{4}$$

$$S_j[t] = \Theta(V_j[t]) \tag{5}$$

where $\hat{t}_i$ is a spike timing of neuron $i$, $\mathcal{T}_{i,j,t} = \{\tau | \hat{t}_j^{\text{last}}[t] < \tau \leq t, S_i[\tau] = 1\}$, and $\hat{t}_j^{\text{last}}[t]$ is the last spike timing of neuron $j$ before $t$. We call this type of description as the **SRM-based description** as it is in the form of the Spike Response Model (SRM) [18] (Figure 1b). Detailed explanations on the equivalence of the two descriptions are given in Appendix B.

## 2.2 Existing gradient computation methods

### 2.2.1 Activation-based methods

To back-propagate the gradients to the lower layers, the activation-based methods [2, 11–13] approximate the derivative of the spiking function which is zero almost everywhere. It is similar to what non-spiking NNs do to the quantized activation functions such as the thresholding function for Binary Neural Networks [19]. The approximated derivative is called the surrogate derivative [14], and we will denote this as $\sigma(V[t]) \approx \frac{\partial S[t]}{\partial V[t]}$.

**RNN-like method**    Since the forward pass of the RNN-like description of the neuron model resembles that of non-spiking RNNs (Figure 1a), back-propagation can also be treated like the Back-Propagation-Through-Time (BPTT) [20] (Figure 2a, the equations are in Appendix C) [2, 12].

**SRM-based method**    However, from the SRM-based description of the same model (Figure 1b), back-propagation is derived in a slightly different way using the kernel function $\varepsilon$ between each layer (Figure 2b) [11]. From Equation 4, we can obtain the gradient of the membrane potential of the postsynaptic neuron $j$ at arbitrary time step $t_a$ with respect to the spike activation of the presynaptic neuron $i$ at time step $t$ as

$$\frac{\partial V_j[t_a]}{\partial S_i[t]} = \begin{cases} w_{i,j} \varepsilon[t_a - t] & \text{if } t > \hat{t}_j^{\text{last}}[t_a] \text{ and } t_a \geq t \\ 0 & \text{else} \end{cases} \tag{6}$$

Interestingly, we found that the SRM-based method (Figure 2b) is functionally equivalent to the RNN-like method except that the diagonal reset paths are removed (Figure 2c, See Appendix D for detailed explanation). In fact, neglecting the reset paths in back-propagation can improve the learning result as it can avoid the accumulation of the approximation errors [21]. Via the reset paths (red dashed arrows in Figure 2a), the same gradient value recursively passes through the surrogate

derivative (red solid arrows in Figure 2a), as many times as the number of time steps. Even though the amount of the approximation error from a single surrogate derivative is tolerable, the accumulated error can be orders of magnitude larger because the number of time steps is usually larger than hundreds. We experimentally observed that propagating gradients via the reset paths significantly degrades training results regardless of the task and network settings. In this regard, we used the SRM-based method instead of the RNN-like method to represent the activation methods throughout this paper.

### 2.2.2 Timing-based methods

The timing-based methods [8–10, 15] exploit the differentiable relationship between the spike timing $\hat{t}$ and the membrane potential at the spike timing $V(\hat{t})$. The local linearity assumption of the membrane potential around $\hat{t}$ leads to $\frac{\partial \hat{t}_i}{\partial V_i(\hat{t}_i)} = -\frac{1}{V_i'(\hat{t}_i)}$ where $V'(t)$ is the time derivative of the membrane potential at time $t$. In this work, we used approximated time derivative $V^*[t] = V[t] - V[t-1]$ for discrete time domain as $\frac{\partial \hat{t}_i}{\partial V_i[\hat{t}_i]} \approx -\frac{1}{V_i^*[\hat{t}_i]}$. Note that computing the gradient of a spike timing does not require the derivative of the spiking function $\Theta$.

From Equation 4 of the SRM-based description, we can obtain the gradient of the membrane potential of the postsynaptic neuron $j$ at arbitrary time step $t_a$ with respect to the spike timing $\hat{t}_i$ of the presynaptic neuron $i$ as

$$\frac{\partial V_j[t_a]}{\partial \hat{t}_i} = \begin{cases} w_{i,j} \frac{\partial \varepsilon[t_a - \hat{t}_i]}{\partial \hat{t}_i} = w_{i,j} \varepsilon^*[t_a - \hat{t}_i] & \text{if } \hat{t}_i > \hat{t}_j^{\text{last}}[t_a] \text{ and } t_a \geq \hat{t}_i \\ 0 & \text{else} \end{cases} \quad (7)$$

where $\varepsilon^*[t]$ is the approximated time derivative of SRM kernel $\varepsilon$ in discrete time domain. Figure 2d depicts how the timing-based method propagates the gradients. Only in the time steps with spikes, $\frac{\partial L}{\partial \hat{t}}$ is propagated to $\frac{\partial L}{\partial V}$ and then is propagated to the lower layer with Equation 7.

Commonly, the timing-based methods limit each neuron to emit at most one spike in their networks. In the multi-spike situation, considering the effect of timing change requires complicated computations due to a recursive effect on subsequent spike timings [22]. However, in the neuron model with the reset in both potential and current, an infinitesimal timing change does not affect subsequent spike timings. It allows us to seamlessly extend the application of the the timing-based methods to multi-spike situations without increasing the computational cost.

## 3 Activation- and Timing-based Learning Rule (ANTLR)

### 3.1 Complementary nature of activation-based methods and timing-based methods

Calculating the gradients is to estimate how much the network output varies when the parameters or the variables are changed. One of the main findings in our study is that the activation-based and timing-based methods are complementary in the way they consider the change in the network.

The change in the spike-train of neuron $i$ can be represented by the generation, the removal, and the shift of spikes. The generation or the removal of a spike is expressed as the change of the spike activation: $\Delta S_i[t] = +1$ for generation and $\Delta S_i[t] = -1$ for removal. The activation-based methods, which calculate the gradient with respect to the spike activations $\frac{\partial L}{\partial S[t]}$, then naturally can consider the generations and the removals. On the other hand, the shift of a spike is expressed as the change of the spike timing: $\Delta \hat{t}_i$ (Figure 3a). The timing-based methods, which calculate the gradient with respect to the spike timings $\frac{\partial L}{\partial \hat{t}}$, easily take account of the spike shifts.

The problem in the activation-based methods is that they cannot deal with the spike shifts accurately. In terms of the spike activations, the spike shift is interpreted as a pair of opposite spike activation changes with causal relationship through the reset path: $\Delta S_i[t_{\text{before}}] = -1, \Delta S_i[t_{\text{after}}] = +1$ (Figure 3b). Because of the major role of the reset path in the spike shift, gradient computation methods with the spike activations cannot consider the shift without precisely computing the gradients related to the reset paths. Unfortunately, as explained in Section 2.2.1, the SRM-based activation-based method does not have a reset path so that it is not possible to consider the spike shift at all. The RNN-like activation-based method has the reset paths, but it suffers from accuracy loss due to the accumulated errors in the reset path. Although the shift of an individual spike does not make a huge difference

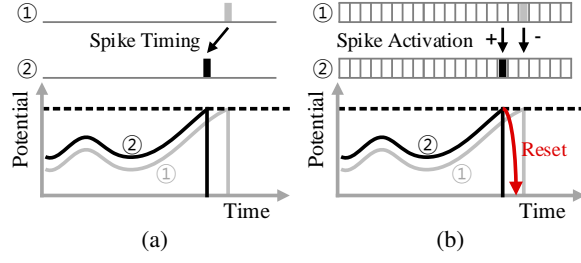

Figure 3: The spike timing shift (①→②) can be described using the change in (a) the spike timing or (b) the spike activation. The spike activation change in the earlier time step causes the activation change in the later time step via the reset path (red arrow).

to the whole network in the situation where many spikes are generated and removed, it becomes important when there are not many spikes in the network.

The problem in the timing-based methods is that the generation and the removal of spikes cannot be described with the spike timings. The timing-based methods also cannot anticipate the spike number change in the network, which happens by the generation or the removal of spikes. Even though the generation and the removal happen less often compared to the spike shift when the parameters are updated by small amounts, their influences to the network are usually more significant. The timing-based methods may unintentionally generate/remove spikes as a result of parameter update while they try to shift the spike timings to reduce the loss, but these unintended generations/removals of spikes do not contribute to training.

## 3.2 Combining activation-based gradients and timing-based gradients

To overcome the limitations in previous works, we propose a new method of back-propagation for SNNs, called an activation- and timing-based learning rule (ANTLR), that combines the activation-based gradients and the timing-based gradients together. The activation-based methods and the timing-based methods back-propagate the gradient through different intermediate gradients, which are $\frac{\partial L}{\partial S}$ and $\frac{\partial L}{\partial t}$, respectively. For this reason, the two approaches have been treated as completely different approaches. However, there is another intermediate gradient $\frac{\partial L}{\partial V}$ calculated in both approaches. $\frac{\partial L}{\partial V}$ in the activation-based methods is propagated from $\frac{\partial L}{\partial S}$ and carries information about the generation and the removal of the spikes whereas $\frac{\partial L}{\partial V}$ in the timing-based methods is propagated from $\frac{\partial L}{\partial t}$ and carries information about the spike shift.

The main idea of ANTLR is to (1) combine the activation-based gradients $\frac{\partial L}{\partial V}|_\text{act}$ and the timing-based gradients $\frac{\partial L}{\partial V}|_\text{tim}$ by taking weighted sum and (2) propagate the combined gradients $\frac{\partial L}{\partial V}|_\text{ant}$ (Figure 4). In ANTLR, the gradients are back-propagated to the lower layers as

$$\frac{\partial L}{\partial V_j[t]}\bigg|_\text{ant} = \lambda_\text{act} \frac{\partial L}{\partial V_j[t]}\bigg|_\text{act} + \lambda_\text{tim} \frac{\partial L}{\partial V_j[t]}\bigg|_\text{tim} \tag{8}$$

$$\frac{\partial L}{\partial V_i[t]}\bigg|_\text{act} = \sum_j \sum_{t_a} \frac{\partial L}{\partial V_j[t_a]}\bigg|_\text{ant} \frac{\partial V_j[t_a]}{\partial S_i[t]} \frac{\partial S_i[t]}{\partial V_i[t]} \tag{9}$$

$$\frac{\partial L}{\partial V_i[\hat{t}_i]}\bigg|_\text{tim} = \sum_j \sum_{t_a} \frac{\partial L}{\partial V_j[t_a]}\bigg|_\text{ant} \frac{\partial V_j[t_a]}{\partial \hat{t}_i} \frac{\partial \hat{t}_i}{\partial V_i[\hat{t}_i]} \tag{10}$$

where last two terms in Equation 9 are calculated using the activation-based method as in Section 2.2.1 and last two terms in Equation 10 are calculated using the timing-based method as in Section 2.2.2. To train SNNs using ANTLR and other methods, we implemented CUDA-compatible gradient computation functions in PyTorch [23] (implementation details[1] are described in Appendix E).

We introduced the coefficients $\lambda_\text{act}, \lambda_\text{tim}$ to balance the gradients from two methods. However, in this work, we used the simplest setting $\lambda_\text{act} = \lambda_\text{tim} = 1$ to focus on showing the fundamental benefits of combining them. Optimal configuration of $\lambda_\text{act}, \lambda_\text{tim}$ should further be studied, as it depends on several

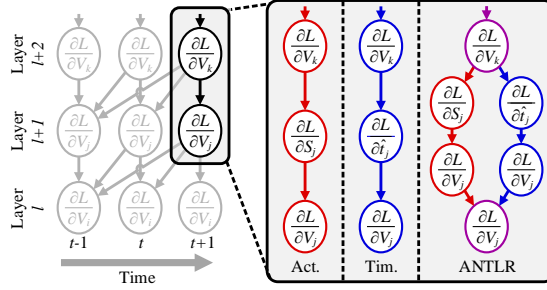

Figure 4: Back-propagation in both the activation-based method and the timing-based method can be described using $\frac{\partial L}{\partial V}$ of neurons at different time steps and the way they are propagated (black arrows). ANTLR combines the two methods (red arrows and blue arrows) by weighted summation at each stage.

| Type | Count | Spike-train | Latency |
|---|---|---|---|
| Loss ($L$) | $\sum_o \{(\sum_\tau S_o[\tau]) - n_o\}^2 / T$ | $\sum_o \sum_\tau d_o[\tau]^2$ | $-\sum_o y_o \log p_o$ |
| $\frac{\partial L}{\partial S_o[t]}$ | $2\{(\sum_\tau S_o[\tau]) - n_o\}/T$ | $2\sum_\tau \kappa[\tau - t]d_o[\tau]$ | $0$ |
| $\frac{\partial L}{\partial \hat{t}_o}$ | $0$ | $-2\sum_\tau \kappa^*[\tau - \hat{t}_o]d_o[\tau]$ | $-\beta(p_o - y_o)$ |
| Compatible with | Activation, ANTLR | Activation, Timing, ANTLR | Timing, ANTLR |

$o$ represents an index of the output neurons, $d_o[\tau] = (\kappa * S_o)[\tau] - (\kappa * S_o^{\text{tar}})[\tau]$, $p_o = e^{-\beta \hat{t}_o^{\text{first}}} / \sum_x e^{-\beta \hat{t}_x^{\text{first}}}$, $\kappa$ represents an exponential kernel, $\beta$ is a scaling factor, $n_o$ represents a target spike number, and $y_o$ represents a target probability

Table 1: Three different types of loss functions and corresponding activation-based gradient $\frac{\partial L}{\partial S_o[t]}$ and timing-based gradient $\frac{\partial L}{\partial \hat{t}_o}$

factors. For example, the scale of the activation-based gradient can be arbitrarily changed by the hyper-parameters of the surrogate derivative, so the optimal configurations can be different depending on such hyper-parameters. Note that ANTLR with the setting $\lambda_{\text{act}} = 1$, $\lambda_{\text{tim}} = 0$ is equivalent to the activation-based method whereas ANTLR with $\lambda_{\text{act}} = 0$, $\lambda_{\text{tim}} = 1$ is equivalent to the timing-based method. Therefore, ANTLR can also be regarded as a unified framework that covers the two distinct approaches.

### 3.3 Loss functions

We used three types of widely used loss functions which are *count* loss, *spike-train* loss, and *latency* loss (Table 1). Count loss is defined as a sum of squared error between the output and target number of spikes of each output neuron. Spike-train loss is a sum of squared error between the filtered output spike-train and the filtered target spike-train. Latency loss is defined as the cross-entropy of the softmax of negatively weighted first spike timings of output neurons. Note that the count loss cannot provide the gradient with respect to the spike timing whereas the latency loss cannot provide the gradient with respect to the spike activation. It makes those loss types inapplicable to certain types of learning methods. We want to emphasize that ANTLR can use all the loss types.

### 3.4 Estimated loss landscape

We conducted a simple experiment to visualize the gradients computed by each method. A fully-connected network with two hidden layers of 10-50-50-1 neurons was trained to minimize the spike-train loss with three random input spikes for each input neuron and a single target spike for the target neuron. After reaching to the global optimum of zero loss, we perturbed all trainable parameters (weights and biases) along first two principal components of the gradient vectors used in training and measured the true loss (Figure 5a). The lowest point at the center (dark blue region) represents the global minimum, and subtle loss increase around the center shows the effect of the spike timing shift. Dramatic increase of the loss depicted in the right corner shows the loss increase from the spike number change. To emphasize the subtle height difference due to the spike timing shift, we highlighted the area adjacent to the global optimum where the number of spikes does not change using the color scheme in Figure 5e.

Different learning methods provide different gradient values based on their distinct approaches. Using each method's gradient vector at each parameter point, we visualized the estimated loss landscape using the surface reconstruction method [24, 25] (Figure 5b to 5d). The results of the activation-based

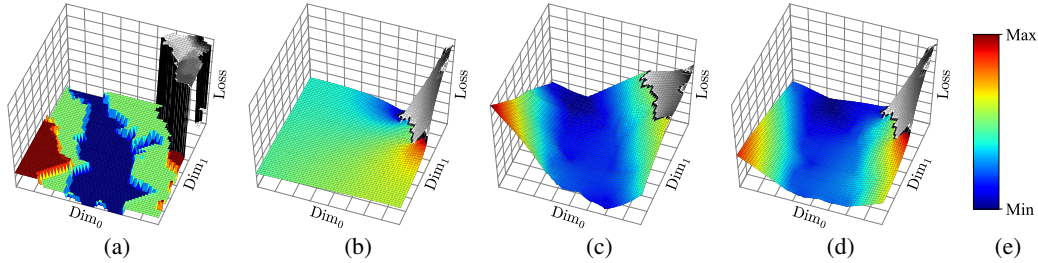

Figure 5: (a) True loss landscape, estimated loss landscapes using (b) the activation-based method, (c) the timing-based method and (d) ANTLR with $\lambda_{act}, \lambda_{tim} = 1, 1$, and (e) the color scheme used for highlighting. $Dim_0$ and $Dim_1$ represent two dimensions along which we perturbed the network parameters.

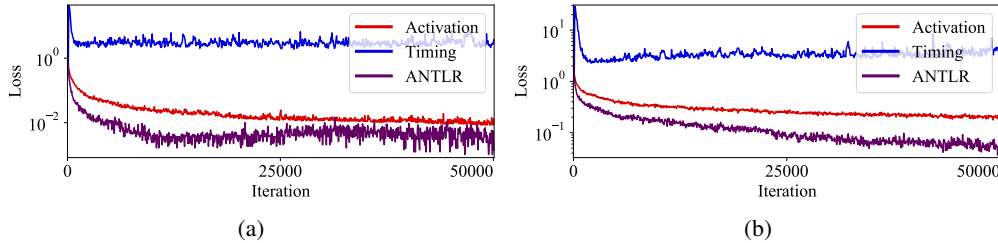

Figure 6: Averaged training loss over 100 trials of random spike-train matching task with three input spikes and (a) a single target spike and (b) three target spikes. Note that the y axis is in logarithmic scale.

method (Figure 5b) well demonstrated the steep loss change due to the spike number change, whereas the timing-based method (Figure 5c) could not take account of it. On the other hand, the timing-based method captured the subtle loss change due to the spike timing shift while the activation-based method showed almost flat loss landscape in the region without the spike number change. By combining both methods, ANTLR was able to capture those features at the same time (Figure 5d).

## 4 Experimental results

We evaluated practical advantages of ANTLR compared to other methods using 3 different tasks: (1) random spike-train matching, (2) latency-coded MNIST, and (3) N-MNIST. Hyper-parameters for training were grid-searched for each task (detailed experimental settings are in Appendix F). Since different training options (e.g. loss type) are available for different learning methods, we tested every option available to each method and reported the best (in terms of accuracy and efficiency) results from each method. The timing-based methods cannot train parameters when a neuron does not emit any spike (dead neuron problem), so we added a no-spike penalty for the timing-based methods that increases the incoming synaptic weights of the neurons without any spike and encourages every neuron to emit at least one spike as in [8].

### 4.1 Random spike-train matching

Using the same experiment setup as in Section 3.4 except the varying number of the target spikes and the different network size of 10-50-50-5, we measured the training loss of the networks trained by different learning methods (Figure 6). This task was used to see the basic performance of the learning methods in a situation where each spike significantly affects the training results. During 50000 training iterations, both the activation-based method and ANTLR showed noticeable decrease in loss whereas the timing-based method failed to train the network as it cannot handle the spike number change. ANTLR outperformed other methods with much faster convergence and lower loss.

### 4.2 Latency-coded MNIST

In this experiment, we applied the latency coding to the input data of MNIST dataset [26] as in [8–10]. The larger intensity value of each pixel was represented by the earlier spike timing of corresponding

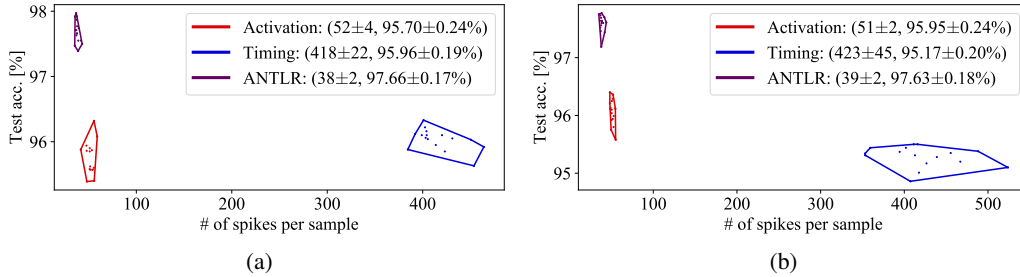

Figure 7: Test accuracy and the required number of hidden and output spikes to classify a single sample on (a) latency-coded MNIST task and (b) latency-coded MNIST task with the single-spike restriction. The values in the legend represent the mean and standard deviation of 16 trials.

input neuron. We used this conversion to reduce the total number of spikes and make the situation where each learning method should take account of the precise spike timing for a better result.

The timing-based method and ANTLR used the latency loss, and the activation-based method used the count loss with the target spike number of 1/0 for correct/wrong labels. To generate at least a single spike for the target output neuron, we added a variant of the count loss $\{\min(\sum_\tau S_d[\tau], 1) - 1\}^2$ ($d$ is the index of the desired class label) to the total loss for ANTLR.

Note that the target spike number for the activation-based method is much smaller than that from previous works since we applied the latency coding to the input to reduce the number of input spikes. The output class can either be determined using the output neuron emitting the most spikes (most-spike decision scheme) or the neuron emitting the earliest spike (earliest-spike decision scheme). The timing-based method and ANTLR used the earliest-spike decision scheme whereas the activation-based method used the most-spike decision scheme considering the loss types they used.

We trained the network with a size of 784-800-10 and 100 time steps using a mini-batch size of 16 and the split of 50000/10000 images for training/validation dataset. The results of test accuracy and the number of spikes used for each sample are shown in Figure 7a. The number of spikes used to finish a task was usually not presented in previous works, but we included it to demonstrate the efficiency of the networks trained by different methods. The results show that ANTLR achieved the highest accuracy compared to other methods. The number of spikes for the timing-based method was exceptionally higher than the others, because of the no-spike penalty that encourages every neuron to emit at least one spike and its inability to remove existing spikes during training. With the help of the activation-based part, ANTLR can add/remove spikes in both hidden and output neurons while allowing some neurons not to emit any spikes. Figure 7b shows a different scenario we tested, where each neuron is restricted to emit at most one spike as in [8–10, 15]. We tested this situation to further reduce the number of spikes. However, this modification did not change the trend of the results as the number of spikes was already small in the first place.

### 4.3 N-MNIST

In contrast to the MNIST dataset which is static, the spiking version of MNIST, called N-MNIST is a dynamic dataset that contains the samples of the input spikes in 34x34 spatial domain with two channels along 300 time steps [27]. The same loss and the classification settings as in Section 4.2 were used here except the target spike number for the activation-based method, which is increased to 10/0 considering the increased number of input spikes in the N-MNIST dataset. Note that the latency loss and the earliest-spike decision scheme have never been used for the N-MNIST dataset, but we intentionally used them to reduce the number of spikes. We trained the network with a size of 2x34x34-800-10 using a mini-batch size of 16 and the results are shown in Figure 8a.

Due to the large target spike number, the activation-based method required much more spikes than ANTLR. The timing-based method again used large number of spikes because of its limitation in removing spikes. We also tested the scenario where the single-spike restriction is applied (Figure 8b). Since the activation-based method had to use the target spike number of 1/0 due to the restriction, its accuracy result was degraded whereas the timing-based method showed improvement in both

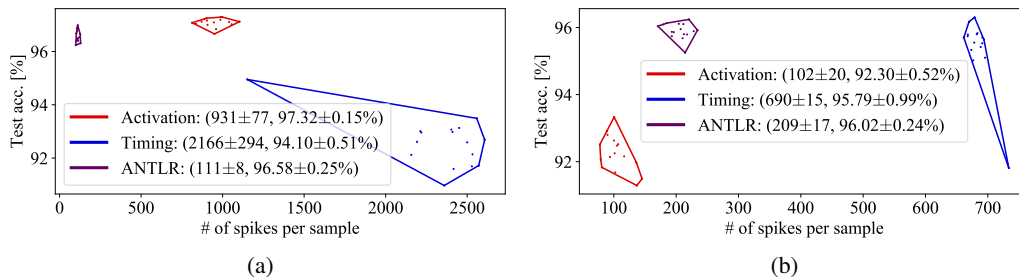

Figure 8: Test accuracy and the required number of hidden and output spikes to classify a single sample on (a) N-MNIST task and (b) N-MNIST task with the single-spike restriction. The values in the legend represent the mean and standard deviation of 16 trials.

accuracy and efficiency. This supports the fact that the activation-based method favors the multi-spike situation and the timing-based method favors the single-spike situation.

Several previous works using the existing learning methods reported higher accuracy than our results, but they did not report exact results of spike numbers. Our experiments using similar settings implied that previous works with high accuracy were benefited from the large number of spikes used (Appendix G). In this study, we focus on the cases in which the networks are forced to use fewer spikes for high energy efficiency. We believe that such cases represent more desirable environments for application of SNNs.

# 5 Discussion and conclusion

In this work, we presented and compared the characteristics of two existing approaches of gradient-based supervised learning methods for SNN and proposed a new learning method called ANTLR that combines them. The experimental results using various tasks showed that the proposed method can improve the accuracy of the network in the situations where the number of spikes are constrained, by precisely considering the influence of individual spikes. Experiments and analysis of ANTLR on larger datasets remain as a future study.

It is known that both the temporal coding and the rate coding play important roles for information processing in biological neurons [28]. Interestingly, the timing-based methods are closely related to the temporal coding since they explicitly consider the spike timings in gradient computation. On the other hand, the activation-based methods are more favorable to the rate coding in which the spike timing change does not contain information. Even though we did not explicitly address the concept of the temporal coding and the rate coding in this work, to the best of our knowledge, this work is the first work that tries to unify the different learning methods suitable for different coding schemes.

Some other works that were not mentioned in this paper also have shown notable results as supervised learning methods for SNNs [29–31], but these methods are not classified as activation-based or timing-based. In these methods, a scalar variable mediates the back-propagation from the whole spike-train of a postsynaptic neuron to the whole spike-train of a presynaptic neuron. This variable may be able to capture the current state of the spike-train and its influence to another neuron, but it cannot cope with the change in the spike-train such as the generation, the removal, or the timing shift during training. This limitation may not be problematic with the rate coding in which the change in the state of individual spikes does not make a huge difference, but it is a critical problem when training SNNs with fewer spikes for higher efficiency.

# Broader Impact

The purpose of our work is to improve the general supervised learning performance of SNNs. Even though we can use SNNs for any cognitive task, the complexity of problems that SNNs are currently targeting is very limited. This is because of the fundamental problems of the existing learning methods that are addressed in this work. Nevertheless, SNNs have significant implications as a biologically plausible artificial neural network, which helps bridge the gap between our understanding

of biological neurons and the remarkable success of deep learning. In particular, the successful use of SNNs can provide clues to the high energy efficiency of the biological brains. We believe our work lays the groundwork for such a research direction.

## Acknowledgments and Disclosure of Funding

This research was supported by Samsung Research Funding Center of Samsung Electronics under Project Number SRFC-TC1603-51, the MSIT (Ministry of Science and ICT), Korea, under the ICT Consilience Creative program (IITP-2019-2011-1-00783) supervised by the IITP (Institute for Information & communications Technology Promotion), and NRF (National Research Foundation of Korea) Grant funded by the Korean Government (NRF-2016-Global Ph.D. Fellowship Program).

## Footnotes

[1] The source code is available at `https://github.com/KyungsuKim42/ANTLR`.

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
