[Supplementary Material]

# Appendix

## A    Versatility of the neuron model

In our neuron model, depending on the decay coefficients $\alpha_V, \alpha_I$, the shape of the post-synaptic potential induced by a single spike can be varied. Figure A1 shows some examples cases of commonly used neuron models that can be implemented using our neuron model.

(a) $\alpha_V, \alpha_I = 1, 0$    (b) $\alpha_V, \alpha_I = 0.95, 0$  (c) $\alpha_V = \alpha_I = 0.95$  (d) $\alpha_V, \alpha_I = 1, 0.95$    (e) $\alpha_V = \alpha_I = 1$

Figure A1: Various types of neuron models can be expressed by the neuron model we used, including (a) simple IF neuron, (b) LIF neuron without decaying synaptic current, (c) biologically-plausible alpha synaptic function [8, 11], (d) non-leaky neuron with exponential PSP [9], and (e) non-leaky neuron with linear PSP [6].

## B    Functional equivalence of the RNN-like description and the SRM-based description of the model

From the RNN-like description of the model (Equation 1 to 3), we can infer that the post-synaptic potential induced by $S_i[t]$, the spike activation of presynaptic neuron $i$ at time step $t$, to $V_j[t_a]$, the potential of a postsynaptic neuron $j$ at later time step $t_a > t$, can be transmitted only via $I_j[t]$. Then $I_j[t]$ forwards the influence to $I_j[t+1]$ and $V_j[t]$, and it continues with $I_j$s and $V_j$s along the way.

If there is no spike activation $S_j[x] = 1$ between $t$ and $t_a$ ($t < x < t_a$), this influence can reach to $V_j[t_a]$, and by the time it reaches, the amount of the influence from $S_i[t]$ becomes $w_{i,j}\beta_I\beta_V \sum_{k=0}^{t_a-t} \alpha_I^k \alpha_V^{t_a-t-k}$. If there is the spike activation $S_j[x] = 1$ between $t$ and $t_a$ ($t < x < t_a$), this influence cannot be transmitted to $V_j[t_a]$ since $S_j[x]$ cuts off the signals that $I_j[x+1]$ and $V_j[x+1]$ receive.

If we express this relationship between $S_i[t]$ and $V_j[t_a]$ with a single kernel function $\varepsilon[\tau] = \beta_I\beta_V \sum_{k=0}^{\tau} \alpha_I^k \alpha_V^{\tau-k}$ and the causal set $\mathcal{T}_{i,j,t} = \{\tau | \hat{t}_j^{\text{last}}[t] < \tau \le t, S_i[\tau] = 1\}$, it becomes the SRM-based description (Equation 4 and 5).

## C    RNN-like activation-based method

From the RNN-like description of the model (Equation 1 to 3), following BPTT-like back-propagation can be derived

$$\frac{\partial L}{\partial V_j[t]} = \frac{\partial L}{\partial S_j[t]}\frac{\partial S_j[t]}{\partial V_j[t]} + \frac{\partial L}{\partial V_j[t+1]}\frac{\partial V_j[t+1]}{\partial V_j[t]} \tag{11}$$

$$\frac{\partial L}{\partial I_j[t]} = \frac{\partial L}{\partial V_j[t]}\frac{\partial V_j[t]}{\partial I_j[t]} + \frac{\partial L}{\partial I_j[t+1]}\frac{\partial I_j[t+1]}{\partial I_j[t]} \tag{12}$$

$$\frac{\partial L}{\partial S_j[t]} = \frac{\partial L}{\partial I_k[t]}\frac{\partial I_k[t]}{\partial S_j[t]} + \frac{\partial L}{\partial I_j[t+1]}\frac{\partial I_j[t+1]}{\partial S_j[t]} + \frac{\partial L}{\partial V_j[t+1]}\frac{\partial V_j[t+1]}{\partial S_j[t]} \tag{13}$$

$$\frac{\partial S_j[t]}{\partial V_j[t]} = \sigma(V_j[t]), \quad \frac{\partial V_j[t+1]}{\partial V_j[t]} = \alpha_V(1 - S_j[t]) \tag{14}$$

$$\frac{\partial V_j[t]}{\partial I_j[t]} = \beta_V, \quad \frac{\partial I_j[t+1]}{\partial I_j[t]} = \alpha_I(1 - S_j[t]) \tag{15}$$

$$\frac{\partial I_k[t]}{\partial S_j[t]} = \beta_I w_{k,j}, \quad \frac{\partial I_j[t+1]}{\partial S_j[t]} = -\alpha_I I_j[t], \quad \frac{\partial V_j[t+1]}{\partial S_j[t]} = -\alpha_V V_j[t] \tag{16}$$

that results in the gradients for the parameter update as

$$\frac{\partial L}{\partial w_{i,j}} = \sum_t \left\{ \frac{\partial L}{\partial I_j[t]} \beta_I S_i[t] \right\}, \quad \frac{\partial L}{\partial V_{\text{bias},j}} = \sum_t \left\{ \frac{\partial L}{\partial V_j[t]} \beta_{\text{bias}} \right\} \tag{17}$$

## D Interpreting SRM-based activation-based back-propagation with RNN-like description

The forward passes of the RNN-like description and the SRM-based description are functionally equivalent, but corresponding back-propagation methods derived from them are slightly different.

The SRM-based back-propagation can be summarized using the relationship between the potentials as follows.

$$\frac{\partial V_j[t_a]}{\partial V_i[t]} = \begin{cases} w_{i,j}\sigma(V_i(t))\varepsilon[t_a - t] & \text{if } t > t_j^{\text{last}}[t_a] \\ 0 & \text{else} \end{cases} \tag{18}$$

where the kernel function is given as $\varepsilon[\tau] = \beta_I \beta_V \sum_{k=0}^{\tau} \alpha_I^k \alpha_V^{\tau-k}$

Similar to the derivation in Appendix B, following back-propagation formula can provide the same functionality as the SRM-based back-propagation.

$$\frac{\partial L}{\partial V_j[t]} = \frac{\partial L}{\partial S_j[t]} \frac{\partial S_j[t]}{\partial V_j[t]} \tag{19}$$

$$\frac{\partial L}{\partial V_j^{\text{dep}}[t]} = \frac{\partial L}{\partial V_j[t]} + \frac{\partial L}{\partial V_j^{\text{dep}}[t+1]} \frac{\partial V_j^{\text{dep}}[t+1]}{\partial V_j^{\text{dep}}[t]} \tag{20}$$

$$\frac{\partial L}{\partial I_j[t]} = \frac{\partial L}{\partial V_j^{\text{dep}}[t]} \frac{\partial V_j^{\text{dep}}[t]}{\partial I_j[t]} + \frac{\partial L}{\partial I_j[t+1]} \frac{\partial I_j[t+1]}{\partial I_j[t]} \tag{21}$$

$$\frac{\partial L}{\partial S_j[t]} = \frac{\partial L}{\partial I_k[t]} \frac{\partial I_k[t]}{\partial S_j[t]} \tag{22}$$

$$\frac{\partial S_j[t]}{\partial V_j[t]} = \sigma(V_j[t]), \quad \frac{\partial V_j^{\text{dep}}[t+1]}{\partial V_j^{\text{dep}}[t]} = \alpha_V(1 - S_j[t]), \tag{23}$$

$$\frac{\partial V_j^{\text{dep}}[t]}{\partial I_j[t]} = \beta_V, \quad \frac{\partial I_j[t+1]}{\partial I_j[t]} = \alpha_I(1 - S_j[t]), \tag{24}$$

$$\frac{\partial I_k[t]}{\partial S_j[t]} = \beta_I w_{k,j} \tag{25}$$

where $V^{\text{dep}}$ is introduced to consider temporal dependency between $V[t]$s of the same neuron at different time steps.

Those formula are almost identical to the RNN-like back-propagation (Equation 11 to 16) except how $\frac{\partial L}{\partial S}$ is propagated (Equation 13 and 22). The only difference is whether the reset paths (red dashed arrows in Figure 2a, represented as $\frac{\partial I_j[t+1]}{\partial S_j[t]}$ and $\frac{\partial V_j[t+1]}{\partial S_j[t]}$) are considered in back-propagation or not.

## E Implementation details of the learning methods

For the activation-based method and ANTLR, we used the surrogate derivative using exponential function $\sigma(v) = \alpha_\sigma \exp(-\beta_\sigma|\theta - v|)$ as in [11]. For the timing-based method and ANTLR, the approximated time derivative $V^*[\tau]$ and $\varepsilon^*[\tau]$ were calculated as $V[\tau] - V[\tau-1]$ and $(\varepsilon[\tau+1] - \varepsilon[\tau-1])/2$ respectively.

Algorithm 1, 2, 3 show the detailed procedure for back-propagation of the activation-based method, the timing-based method, and ANTLR, respectively; $\frac{\partial L}{\partial X}$ is represented as $\delta X$ for better readability, and $W^l$ represents a weight matrix between layer $l$ and layer $l + 1$. Note that $\frac{\partial L}{\partial S}[t]$ and $\frac{\partial L}{\partial \hat{t}}[t]$ are calculated considering the loss function used (Table 1). $V_{\text{dep}}$ from Appendix D was used in all

methods to reduce the total number of computations by not using $\varepsilon$ explicitly. For the same reason, we did not implement the **for** loop related to $\varepsilon^*$ (Algorithm 2 and 3) in the actual implementation and used auxiliary variables similar to $V_{\text{dep}}$.

---

**Algorithm 1:** The activation-based back-propagation

---

**for** $t = T - 1$ **to** $0$ **do**
    **for** $l = L - 1$ **to** $0$ **do**
        **if** $l = L - 1$ **then**
            $\delta S^l[t] \leftarrow \frac{\partial L}{\partial S_o}[t]$;
        **else**
            $\delta S^l[t] \leftarrow \sum W^l \delta I^{l+1}[t]$;
        **end**
        $\delta V^l[t] \leftarrow \sigma(V^l[t])\delta S^l[t]$;
        $\delta V^l_{\text{dep}}[t] \leftarrow \delta V^l[t] + \alpha_V(1 - S^l[t])\delta V^l_{\text{dep}}[t+1]$;
        $\delta I^l[t] \leftarrow \beta_V \delta V^l_{\text{dep}}[t] + \alpha_I(1 - S^l[t])\delta I^l[t+1]$;
    **end**
**end**

---

**Algorithm 2:** The timing-based back-propagation

---

**for** $t = T - 1$ **to** $0$ **do**
    **for** $l = L - 1$ **to** $0$ **do**
        **if** $l = L - 1$ **then**
            $\delta \hat{t}^l[t] \leftarrow \frac{\partial L}{\partial \hat{t}_o}[t]$;
        **else**
            **for** $\tau = -1$ **to** $T - t + 1$ **do**
                $\delta \hat{t}^l[t] \leftarrow \delta \hat{t}^l[t] + \sum W^l \varepsilon^*[\tau]\delta V^{l+1}[t+\tau])$;
            **end**
        **end**
        **if** $S^l[t] = 1$ **then**
            $\delta V^l[t] \leftarrow -\delta \hat{t}^l[t]/V^{l*}[t]$;
        **else**
            $\delta V^l[t] \leftarrow 0$;
        **end**
    **end**
**end**

---

**Algorithm 3:** ANTLR back-propagation

---

**for** $t = T - 1$ **to** $0$ **do**
    **for** $l = L - 1$ **to** $0$ **do**
        **if** $l = L - 1$ **then**
            $\delta S^l[t] \leftarrow \frac{\partial L}{\partial S_o}[t]$;
            $\delta \hat{t}^l[t] \leftarrow \frac{\partial L}{\partial \hat{t}_o}[t]$;
        **else**
            $\delta S^l[t] \leftarrow \sum W^l \delta I^{l+1}[t]$;
            **for** $\tau = -1$ **to** $T - t + 1$ **do**
                $\delta \hat{t}^l[t] \leftarrow \delta \hat{t}^l[t] + \sum W^l \varepsilon^*[\tau]\delta V^{l+1}[t+\tau])$;
            **end**
        **end**
        $\delta V^l[t] \leftarrow \lambda_{\text{act}}\sigma(V^l[t])\delta S^l[t]$;
        **if** $S^l[t] = 1$ **then**
            $\delta V^l[t] \leftarrow \delta V^l[t] - \lambda_{\text{tim}}\delta \hat{t}^l[t]/V^{l*}[t]$;
        **end**
        $\delta V^l_{\text{dep}}[t] \leftarrow \delta V^l[t] + \alpha_V(1 - S^l[t])\delta V^l_{\text{dep}}[t+1]$;
        $\delta I^l[t] \leftarrow \beta_V \delta V^l_{\text{dep}}[t] + \alpha_I(1 - S^l[t])\delta I^l[t+1]$;
    **end**
**end**

---

# F Experimental settings

Hyper-parameters used for loss landscape estimation (Section 3.4) and random spike-train matching task (Section 4.1) are listed in Table A1. For latency-coded MNIST task and N-MNIST task, we grid-searched several hyper-parameter options and reported the results of the ones that provided highest valid accuracy (averaged over 16 trials). Table A2 and Table A3 show searched hyper-parameter options and the ones used for the final results.

Some of the hyper-parameters were not mentioned in the paper. `grad_clip` is for clipping the parameter gradients before update. `init_bias_center` was used as a binary option that initialize the bias with large value to ease the generation of spikes at earlier training iterations. `kappa_exp` is for the exponential filter used for the spike-train loss. `ste_alpha` and `ste_beta` are coefficients for the surrogate derivative described in Appendix E.

| Name | Value |
|---|---|
| `alpha_v`, `alpha_i` | 0.95, 0.95 |
| `grad_clip` | 1e5 |
| `init_bias_center` | 0 |
| `kappa_exp` | 0.95 |
| `learning_rate` | 1e-3 |
| `optimizer` | 'sgd' |
| `ste_alpha` | 0.3 |
| `ste_beta` | 1 |

Table A1: Hyper-parameters used for loss landscape estimation (Section 3.4) and random spike-train matching task (Section 4.1)

| Hyper-parameter | Searched options | Chosen for | | |
|---|---|---|---|---|
| | | Activation | Timing | ANTLR |
| `alpha_v`, `alpha_i` | (0.95, 0.95), (0.99, 0.99) | (0.99, 0.99) | (0.99, 0.99) | (0.99, 0.99) |
| `beta_softmax` | 0.5, 1, 2 | - | 1 | 1 |
| `epoch` | 10 | 10 | 10 | 10 |
| `grad_clip` | 1e6, 10, 1 | 1e6 | 1e6 | 1e6 |
| `init_bias_center` | 0, 1 | 0 | 1 | 1 |
| `learning_rate` | 1e-2, 1e-3, 1e-4 | 1e-3 | 1e-4 | 1e-3 |
| `max_target_spikes` | 1 | 1 | - | - |
| `optimizer` | 'adam' | 'adam' | 'adam' | 'adam' |
| `ste_alpha` | 0.3, 1 | 1 | - | 1 |
| `ste_beta` | 1, 3 | 3 | - | 3 |
| `weight_decay` | 0, 1e-3, 1e-4 | 0 | 0 | 0 |

Table A2: Hyper-parameters searched and chosen for latency-coded MNIST task (Section 4.2)

| Hyper-parameter | Searched options | Chosen for | | |
|---|---|---|---|---|
| | | Activation | Timing | ANTLR |
| `alpha_v`, `alpha_i` | (0.95, 0.95), (0.99, 0.99) | (0.99, 0.99) | (0.99, 0.99) | (0.99, 0.99) |
| `beta_softmax` | 1/6, 1/3, 2/3 | - | 1/3 (1/6*) | 1/6 |
| `epoch` | 5 | 5 | 5 | 5 |
| `grad_clip` | 1e6, 10, 1 | 10 (1*) | 1 | 1 |
| `init_bias_center` | 0 | 0 | 0 | 0 |
| `learning_rate` | 1e-2, 1e-3, 1e-4 | 1e-3 | 1e-4 | 1e-3 |
| `max_target_spikes` | 1, 3, 10 (1*) | 10 (1*) | - | - |
| `optimizer` | 'adam' | 'adam' | 'adam' | 'adam' |
| `ste_alpha` | 0.3, 1 | 1 | - | 1 |
| `ste_beta` | 1, 3 | 3 | - | 3 |
| `weight_decay` | 0, 1e-3, 1e-4 | 0 | 0 | 0 |

Table A3: Hyper-parameters searched and chosen for N-MNIST task (*hyper-parameters used in the case with the single-spike coding if they are different) (Section 4.3)

# G Experiments with a higher number of target spikes

Even for the same type of learning method, different experiment settings such as the number of target spike can change the accuracy and the number of spikes results. We compared previous results of fully-connected SNNs on N-MNIST classification tasks with our experimental results with different number of target spikes (Table A4).

Table A4: Comparison of fully-connected SNNs on N-MNIST

| Method | Type* | Test Accuracy [%] | Loss** | # Target spikes | # Spikes/sample |
|---|---|---|---|---|---|
| Lee et al. [27] | S | 98.66 | C | not fixed | N/A |
| Jin et al. [26] | S | 98.84±0.02 | C | 35 / 5 | N/A |
| SLAYER [10] | A | 98.89±0.06 | C | 60 / 10 | N/A |
| STBP [11] | A | 98.78 | C | 300 / 0 | N/A |
| SRM-based*** | A | 97.73±0.14 | C | 10 / 0 | 436±17 |
| ANTLR | A&T | 97.73±0.09 | C | 10 / 0 | 415±14 |
| SRM-based*** | A | 98.30±0.06 | C | 60 / 10 | 6536±120 |
| ANTLR | A&T | 98.05±0.10 | C | 60 / 10 | 6638±130 |

* A (activation-based), T (timing-based), and S (scalar-mediated, refer to Section 5), ** C (count loss) and L (latency loss), *** Our implementation of existing approaches

Compared to 10/0 target spike numbers that we used in this work, using more target spikes can improve the accuracy result but also increases the number of spikes used for inference. It implies that previous works with high accuracy were benefited from the large amount of spike usage. Therefore, it is not fair to compare learning methods solely by accuracy results without considering the number of spikes. Note that accuracy result of ANTLR is not better than the activation-based method when a lot of spikes are used and the timing of individual spike carries almost no information.