[Reviews · NeurIPS 2020]

Review 1

Summary and Contributions: 1) summary: The authors propose a gradient-based supervised learning method called activation- and timing- based learning rule (ANTLR), that can improve the accuracy of the network in the situations where the number of spikes are constrained, by precisely considering the influence of individual spikes. 2) contributions: The key idea is unifying the different learning methods (activation-based and timing-based) suitable for different coding schemes (temporal coding and rate coding). The authors run a series of experiments which demonstrate that the proposed method shows higher performance in terms of both accuracy and efficiency than the activation- based and timing-based approaches.

Strengths: The proposed approach is interesting, since it suggests a way to process patterns that are encoded with both a rate and a temporal code. However, the method proposed in this paper is only integrates the two learning methods together, which is not innovative enough. As the experiments and theoretical analysis are still sufficient and properly assess some of the points made by the author, I think that the paper is a not bad contribution for NeurIPS.

Weaknesses: The method proposed in this paper is only integrates the two learning methods together, which is not innovative enough.

Correctness: In the third paragraph of section 4.2, the author mentions that "The number of spikes used to finish a task was usually not presented in previous works, .... " However, many learning algorithms based on the number of spikes have been proposed. It is suggested that the author investigate relevant studies, correct the expressions, add comparison experiments of Activation-based, ANTLR-based, and other spikes-based methods or analyze their differences, and point out the advantages of ANTLR.

Clarity: The paper is clearly written and easy to follow.

Relation to Prior Work: This paper has clearly discussed how ANTLR differs from previous contributions. And it described how to combine the two learning methods in detail.

Reproducibility: Yes

Additional Feedback: Some issues: -4.1, How to realize the method of adding the no-spike penalty is not clear, please add. -4.2, The author points that ’We also added a variant of the count loss to the total loss of ANTLR to prevent the target output neuron from being silent.’ Please add the formula for the variant loss function. -It is suggested to increase another classification experiment that based on the rate coding and to verify that the ANTLR is suitable for the rate coding data. -Because the network structure has great influence on the number of spikes, it is suggested that the influence of the structure on the network performance is discussed. /////////////////////////////////// Update comments after rebuttal: I thank the authors for their detailed rebuttal letter, which has addressed some of the major concerns. I appreciate the claimed contribution, in the sense that it unifies the different learning methods (activation-based and timing-based) suitable for different coding schemes (temporal coding and rate coding). On the other hand, it just combines those two training approaches. In addition, the parameters used for balancing the gradients have a great influence on the effect of the fusion algorithm, but the authors do not make a detailed analysis, which makes the experimental results not really convincing. So the paper right at the borderline, and a resubmission with more thorough experimental validation would make the work better appreciated.


Review 2

Summary and Contributions: This paper presents a backpropagation (bp) training method (ANTLR) for spiking neural networks by averaging two different bp formulations in the literature in the form of a weighted sum. Essentially, a weighted-sum of the gradients computed by the two well known SNN bp formulations, BPTT with surrogate derivative approximation, referred to as activation-based methods by the authors, and spikepop type bp methods, referred to as timing-based methods in the paper, is empirically used to update the weight/bias parameters of the network. The authors claim this weighted average of the two known formulation offers a better solution to the supervised training of spiking neural networks. The following assessment is based on reading the paper and also submitted author rebuttal. This reviewer is concerned with several major problems of this paper: (P1 lack of novelty): The proposed ANTLR essentially computes the error gradients by averaging the gradients computed by two well-known bp formulations. As such, the presented work has no fundamental new contribution. (P2 lack of mathematical rigor): The averaging scheme in ANTLR is ad hoc, and is suggested with no firm mathematical foundation. It is merely based upon the intuition that having a weighted sum of the two methods can lead to a method can outperform both methods. (P3 poor accuracy of the proposed method): ANTLR is only tested using two small datasets: latency-coded MNIST and NMNIST on small feedforward spiking neural networks with only a single hidden layer. And yet, the reported ANTLR accuracies are obviously worse than other reported methods cited in the paper. (P4 lack of convincing experimental evidence): The authors made no direct quantitative comparison with other published methods in terms of accuracy and sparsity. The experimental settings are inconsistent among the three types of bp methods implemented by the authors and hence are insufficient to support the claimed merits of ANTLR such as sparsity.

Strengths: This paper is easy to follow and written clearly. The authors did a good job providing a survey on the existing activation-based (BPTT with surrogative derivative) and timing-based (SpikePop like) methods, contrasting the differences in their formulations.

Weaknesses: More detailed discussions about the main weaknesses of this work: (P1 lack of novelty): The authors’ main argument is that the activation and timing-based methods have their respective pros and cons, so combining them using a weighted sum of the two (in terms of the intermediate derivative partial_L/partial_V both methods compute) will retain the best of the two worlds. While this is a reasonable assumption, but the idea lacks fundamental new contribution. (P2 lack of mathematical rigor): The weighted-averaging scheme in ANTLR is ad hoc in the sense that it is not derived over a rigorous mathematical basis. Timing or spiking activation are just two facets of the same spiking phenomena. On what basis can the derivatives with respect to timing and activation be added together? I don’t see an appropriate unifying mathematical handling here. The presented averaging scheme is partially contingent upon the argument that activation-based bp methods can add/remove spikes while timing-based methods cannot. In fact, these methods are not mathematically formulated to explicitly add/remove spikes but in practice they can. The same can be said for timing-based methods if the constraint of single spike per neuron is relaxed. Furthermore, there could be different ways of averaging the two gradients. I suspect the presented ANTLR has little difference from the following trivial scheme: run an activation-based and timing-based method completely separately, and in the end using the weighted-sum of the gradients computed independently by the two methods to update the weights/biases. The difference is that the presented ANTLR scheme performs averaging during the BP process for a common internal partial derivative (partial_L/partial_V). This may alter the numerical results somewhat, but the spirit remains the same. As expected, there are practical challenges directly resulted from the ad hoc nature of the approach. How do you choose the weights for the averaging? The authors use equal weights. Why is this optimal? (P3 poor accuracy of the proposed method): ANTLR is only tested using two small datasets: latency-coded MNIST and NMNIST based on very small feedforward spiking neural networks with only a single hidden layer. The authors are encouraged to use much large networks and more challenging datasets to verify the proposed method. The authors didn’t provide any quantitative accuracy comparison with related methods in the literature, which shall be explicitly added to the paper for fair performance analysis. For the latency-coded MNIST (section 4.2), the authors only mentioned that previous works reported higher accuracy, which is indeed the case. On N-MNIST, the authors reported 96.58% accuracy without using the single spike restriction. However, methods published a few years ago like [11, 26] show obviously better results, e.g. the accuracy of [11] is ~99%. The accuracy of ANTLR is not competitive in comparison with other related bp methods. (P4 lack of convincing experimental evidence): while lacking quantitative comparison with other published methods, the authors also used insistent experimental setups to compare their implementations of activation-based, timing-based and ANTLR methods. For example, on the latency-coded MNIST, particularly chosen target spike numbers like 1/0 for correct and incorrect labels are used for the activation-based method. While both ANTLR and timing-based methods are set up with the latency loss, an additional firing count loss term is used for ANTLR. But my suspicion is that adding this addition firing count control will favor ANTLR because it can improve sparsity of firing activities. Furthermore, different decision schemes are used. I am curious why the time-based method leads to a lot more spikes than ANTLR even when the inputs are coded very sparsely using latency. Is this still the case if the weight initialization is done in such a way that the network starts from low-activity? It is mentioned that the timing-based methods produces a lot more spikes than the other two methods because of the no-spike penalty. What is exactly this no-spike penalty? Was it also added to ANTLR for fair comparison? My feeling is that due to the different settings adopted for the three methods, what we see in the results are not necessarily the differences in these methods’ core training performance. At least, activation-based methods and ANTLR can be compared under a common loss function like ones based on firing count and output spike trains since they can both handle such loss functions. The same thing should be done to compare ANTLR with the timing-based methods apple to apple. Furthermore, there is no quantitative comparison with other published methods in terms of network sparsity. The argument that ANTLR produces significantly improved sparsity is not well supported.

Correctness: The presented ANTLR is falling behind with other published methods on accuracy. Due to inconsistent experimental settings used for the three methods implemented by the authors, the relative merits of ANTLR is not well supported by the presented experimental evidence. In addition, the adopted datasets and network sizes are very small; the authors should scale up their experimental effort. These issues are detailed under “Weaknesses”.

Clarity: In general, the paper is easy to follow, particularly on the survey of the existing bp methods. The authors call the particular type of activation-based methods in section 2.1 RNN-like method. “RNN” is a bad wording, and can easily introduce confusion with recurrent networks. Section 2.2.1: it is mentioned that neglecting reset paths can improve the performance. How stable is this conclusion in practice? Are you able to have better performance if reset paths are only considered over a short time window? Figure 3(b) is pretty confusing. What do you mean by a timing shift can be realized by having two spike activations with opposite directions? How can you leverage this in activation-based bp methods to mimic spike timing shifs? If so, how do you control the amount of shift? In ANTLR, can the activation-based bp component be either the RNN-like and SRM based method? I assume the answer is yes.

Relation to Prior Work: Since not all bp methods fall under the categories of the so-called activation/timing based methods, such exceptions shall be made clear upfront in the introduction. As discussed in “Weaknesses”, the authors shall add direct comparison with related bp methods in recent literature experimentally.

Reproducibility: Yes

Additional Feedback:


Review 3

Summary and Contributions: The paper investigates gradient-based supervised learning rules for spiking neural networks (SNNs), in particular those calculating gradients with respect to spike activations, and those calculating gradients with respect to spike timing. A novel method is presented that combines the two paradigms into the ANTLR learning rule, which uses a weighted combination of the two gradients, and is able to deal with loss functions and scenarios that are hard or even infeasible with one or the other approach. The method is tested on spike-train matching, MNIST, and N-MNIST, achieving OK accuracy but with very few spikes.

Strengths: 1. The significance of the paper comes from the integration of two concepts for training SNNs (activation-based gradients and timing-based gradients) that so far have been studied separately. The combination provides advantages that cannot be achieved with only one of the approaches. 2. The ANTLR method is well motivated and clearly derived from existing methods. The originality and novelty is limited though, as its merely a combination of existing building blocks. 3. The presented method is efficient in its use of spikes and achives satisfactory results with a very small spike count (although accuracies are significantly below state-of-the-art approaches for SNNs that use more spikes). 4. Although the evaluation is not done on real-world tasks, it makes use of toy datasets (random spike train matching), classical machine learning tasks (MNIST), and specific spiking datasets (N-MNIST). It thereby shows its usefulness in a variety of relevant tasks for SNNs. 5. The experiments are sufficiently documented to be reproducible, code will be released at a later stage.

Weaknesses: 1. The paper overall lacks clarity, in particular the figures of computational graphs (Fig. 1, 2, 4) are not well explained and lack meaningful captions. 2. The demonstrated advantages are mainly relative to either purely activation or purely timing-based rules, but do not compare to state-of-the-art methods. The accuracies in MNIST and N-MNIST are well below those reached in other papers. The paper admits this and highlights the efficient use of very few spikes. While this is an advantage, I would like to see a recommendation or even better evidence that the accuracy gap can be closed, e.g. with larger networks. 3. Broader impact is not addressed.

Correctness: Yes, the methods are derived and evaluations seem to be done properly.

Clarity: Clarity is average at best, mainly because the figures are hard to interpret and not properly explained in the captions. Some important derivations are made in the appendix only.

Relation to Prior Work: The paper properly cites sources from both the activation- and the timing-based gradient literature for SNNs. There are certainly other relevant approaches that could be cited, but I consider it sufficient.

Reproducibility: Yes

Additional Feedback: I consider this paper relevant work by integrating two previously separate strands of research on SNN training. The results are not really convincing though, since the accuracies are relatively far from the best results achieved with competing approaches. I would recommend adding a comparison to the state-of-the-art, and make proposals how the gap can be overcome. I would strongly recommend making the figures more self-explanatory, e.g. by a longer caption. =================================== Update after rebuttal: I thank the authors for their clear rebuttal letter, which has addressed some of the major concerns. I still think that this is a valuable contribution, since it offers a new perspective on how to deal with the two seemingly opposite views on spike codes. On the other hand, it is slightly ad hoc to just linearly weight the two gradients, and the experimental results are not really convincing. For a conference like NeurIPS this puts the paper right at the borderline, and a resubmission with more thorough experimental validation of the method would sound like the best possibility to me.


Review 4

Summary and Contributions: This paper first discusses the deficiencies of two independent approaches to learning in spiking neural networks: activation-based and timing-based methods. The former considers learning as a process of generating and removing spikes, while the latter considers the shifting of spiking times. Then the paper proposes to combine both approaches in the learning rule. Experimental tests show that the proposed method is both accurate and efficient (in terms of the number of spikes).

Strengths: The proposed method is based on a sound analysis of spike dynamics. The generation, removal and shifting of spikes should all be incorporated in the analysis. Not only is this paper important in proposing a new algorithm, but it also introduces a comprehensive viewpoint that may have long-term impact in the field.

Weaknesses: I queried why the experimental settings are not completely consistent among the three types of backpropagation methods. Nevertheless, the timing-based method and ANTLR seem to have more similar settings and can be compared, and the results showed that ANTLR outperforms the timing-based method in Figs. 7(a), 7(b) and 8(a). For Fig. 8(b), ANTLR and the timing-based method have comparable performance, but there are more spikes in the latter method. Although it was claimed that the no-spike penalty is one of the reasons, it remains for the manuscript to clarify whether including this penalty upsets the fairness of comparison. In the discussion and conclusion section, the manuscript mentioned a few other works that are neither activation-based nor timing-based. To ensure that the supporting evidence of the proposed method is complete, there should be a check to verify whether those methods are also efficient in terms of sparsity. Update after feedback and discussions: ============================= While it is true that the two component algorithms are not new, the manuscript’s contribution is more than merely combining two common algorithms using a weighted sum to retain the best of both worlds. In the combination, the activation-based component has not taken into account the time shift of the spikes, whereas the timing-based component has not taken into account the generation and removal of spikes. The algorithm is therefore combining the two methods with complementary nature. Thus, the approach is considered to be a principled one. I consider this a conceptual advance that will influence future analyses of spike dynamics in a holistic way. Indeed, it is noted that both amplitude and timing characterize a spike. Thus, learning in spiking neural networks should have handles to adjust both elements. Including the gradients due to both activation and time shifts with equal weighting is consistent with the calculus of the chain rule for calculating derivatives of multivariable functions. Although the reported accuracies are worse than other reported methods cited in the manuscript, the focus of this paper was cases in which the networks are forced to use fewer spikes, which are constrained to have lower performance. The manuscript considered latency coding, which belong to the regime of reduced input spikes and is not so much the focus of previous works. On the other hand, I agree with my fellow reviewers that more experimental results are needed to verify the effectiveness of the algorithm. Overall, my score is tuned from 8 to 7.

Correctness: The paper may need to clarify whether the derivative of the membrane potential in Eq. (7) should also be dependent on the last spike of the same neuron, that is, $\tilde t_j^{\rm last}$.

Clarity: Overall, the paper is clearly written, except for a few points to be clarified. Figure 2: Not all partial derivatives appearing in Appendices C and D are displayed in the figure. It may be instructive to mention them in the figure caption. Line 99 and Equation (7): Please explain what approximation is used in the derivative of the membrane potential. Section 4.2: The reader may wonder why different loss functions were used in the three methods to be compared, and whether the inclusion of no-spike penalty upsets the fairness of comparison. Line 221: “No-spike penalty” was applied in the timing-based method, but in the previous paragraph an additional count loss was applied to ANTLR instead. Please clarify.

Relation to Prior Work: The paper discussed how its proposed method was related to the previous activation-based and timing-based methods and how it improves over them.

Reproducibility: Yes

Additional Feedback:

[Author Response · NeurIPS 2020]

Thanks for the constructive comments. Let us categorize them into groups and answer due to the page limit.

**Reasoning behind ANTLR:** We would like to emphasize that our main contribution is to discover the complementary

nature of two main approaches for SNN training and provide a justification to combine them, not just randomly adding

two different gradients. A spike-train of the neuron $i$ can be represented in two different ways: $\Phi_i = \{S_i[t]$ for $t =$

$0, 1, \cdots, T-1\}$ (*activation-based representation*) and $\Phi_i = \{\hat{t}_{i,k}$ for $k = 0, 1, \cdots, N-1\}$ (*timing-based representation*).

One of the main arguments of our paper (in Section 3.1) is that the change in the state of neuron $i$ can be described as

$\Delta\Phi_i = \Delta\Phi_{i,\text{gen}} + \Delta\Phi_{i,\text{remov}} + \Delta\Phi_{i,\text{shift}}$, where $\Delta\Phi_i = \{\Delta S_i[t]$ for $t = 0, 1, \cdots, T-1\}$, $\Delta\Phi_{i,\text{gen}} = \{\Delta S_i[t] = +1$ for $t \in \mathcal{T}_{\text{gen}}\}$,

$\Delta\Phi_{i,\text{remov}} = \{\Delta S_i[t] = -1$ for $t \in \mathcal{T}_{\text{remov}}\}$, and $\Delta\Phi_{i,\text{shift}} = \{\Delta S_i[t_{\text{before}}] = -1, \Delta S_i[t_{\text{after}}] = +1$ for $\{t_{\text{before}}, t_{\text{after}}\} \in \mathcal{T}_{\text{shift}}\} =$

$\{\Delta\hat{t}_{i,k}$ for $k = 0, 1, \cdots, N-1\}$. To update any parameter $x$ for desired network output, $\frac{\partial\Phi_i}{\partial x}$ must be precisely

computed so that it satisfies $\Delta x \cdot \frac{\partial\Phi_i}{\partial x} \approx \Delta\Phi_i$. Due to the limitation in considering the reset path for gradient

computation, $\frac{\Delta\Phi_{i,\text{shift}}}{\Delta x}$ cannot be precisely estimated/predicted by the activation-based gradient $\frac{\partial\Phi_i}{\partial x}\big|_{\text{act}}$. In contrast,

the timing-based gradient $\frac{\partial\Phi_i}{\partial x}\big|_{\text{tim}}$ cannot estimate $\Delta\Phi_{i,\text{gen}}$ and $\Delta\Phi_{i,\text{remov}}$. Those changes cannot even be described

in the *timing-based representation*. The solution we proposed with ANTLR is to compensate the inaccuracy in

gradients by adding two imperfect and complementary values as: $\frac{\partial\Phi_i}{\partial x}\big|_{\text{act}} \approx \frac{\Delta\Phi_{i,\text{gen}}}{\Delta x} + \frac{\Delta\Phi_{i,\text{remov}}}{\Delta x}$, $\frac{\partial\Phi_i}{\partial x}\big|_{\text{tim}} \approx \frac{\Delta\Phi_{i,\text{shift}}}{\Delta x}$,

$\frac{\partial\Phi_i}{\partial x}\big|_{\text{ant}} = \lambda_{\text{act}}\frac{\partial\Phi_i}{\partial x}\big|_{\text{act}} + \lambda_{\text{tim}}\frac{\partial\Phi_i}{\partial x}\big|_{\text{tim}} \approx \lambda_{\text{act}}\frac{\Delta\Phi_{i,\text{gen}}}{\Delta x} + \lambda_{\text{act}}\frac{\Delta\Phi_{i,\text{remov}}}{\Delta x} + \lambda_{\text{tim}}\frac{\Delta\Phi_{i,\text{shift}}}{\Delta x}$

**Layer-wise *vs* network-wise summation:** As Reviewer #2 pointed out, one can run the activation-based method and

the timing-based method separately and use the summation of those two gradients for parameter update. However, there

are major differences between that approach and ANTLR. First of all, depending on the type of the loss function, one

of learning methods may provide zero gradients for every parameter (Table 1). Then combining the two methods by

network-wise summation can be meaningless. Moreover, if the activation- and timing-based gradients are not combined

in each layer, errors in the gradients are accumulated through the back-propagation along multiple layers.

**Coefficients $\lambda$:** We introduced the coefficients $\lambda_{\text{act}}, \lambda_{\text{tim}}$ to balance the gradients from two methods because the scale

of the activation-based gradient can be arbitrarily changed by the hyper-parameters of the surrogate derivative. Even

though we simply used $\lambda_{\text{act}}, \lambda_{\text{tim}} = 1$ in this work for convenience, the optimal coefficients should further be studied.

**Adding/removing spikes:** Reviewer #2 mentioned that timing-based methods can also add/remove spikes when a single

spike constraint is relaxed. Timing-based methods may unintentionally generate/remove spikes as a result of parameter

update while they try to shift the spike timing to reduce the loss. However, these *unintended* generation/removal of

spikes do not contribute to training, as the timing-based method cannot estimate them in gradient computation.

**No-spike penalty:** The no-spike penalty (Line 191-192) has been used in the timing-based methods because they cannot

train parameters when a neuron does not emit any spike (dead neuron problem). It is implemented by encouraging

*every neuron* to emit at least one spike. One of the main advantages of ANTLR is that it can solve the dead neuron

problem more efficiently, using variant of the count loss $\{\min(\sum_\tau S_d[\tau], 1) - 1\}^2$ ($d$ is the index of the desired class

label) that penalizes only *output neurons* without no spike. With the help of the activation-based part, ANTLR can

add/remove spikes in both hidden and output neurons while allowing some neurons not to emit any spikes. Reviewer #2

commented that addition of the count loss would decrease the firing activity but it actually *increases* the spiking activity

because it still penalizes some neurons without spikes.

**Comparison with related works:** In the submission, we did not report the comparison with other works which did

not report the sparsity of spikes since it is not fair to compare each method solely by accuracy results. The accuracy

of SNNs highly depends on the number of spikes used. To support the argument, we compared previous results of

fully-connected SNNs on N-MNIST classification tasks with our experimental results (Table R1). Even though related

works did not report exact results of spike numbers, our experiments using similar settings imply that previous works

with high accuracy were benefited from the large amount of spike usage.

**Experimental Settings:** As the reviewers men-
tioned, our experimental results used different set-
tings for each method. We reported the best (in
terms of accuracy and efficiency) results from each
method after we tested every option available to
each method. ANTLR can provide similar ac-
curacy compared to the activation-based method
when the same setting is used (Table R1). However,
this situation is not desirable for efficient use of
SNNs because it requires larger number of spikes.

**Experiments using larger datasets:** We agree
that our experimental results are only from rel-
atively small datasets. We wanted to focus on fun-
damental limitations of existing learning methods
in gradient computation in this work. Experiments and analysis of ANTLR on larger datasets remain as a future study.

Table R1: Comparison of fully-connected SNNs on N-MNIST

| Method | Type* | Accuracy [%] | Loss** | # Target spikes | # Spikes/sample |
|---|---|---|---|---|---|
| Lee et al. [27] | S | 98.66 | C | not fixed | N/A |
| Jin et al. [26] | S | 98.84±0.02 | C | 35 / 5 | N/A |
| SLAYER [10] | A | 98.89±0.06 | C | 60 / 10 | N/A |
| STBP [11] | A | 98.78 | C | 300 / 0 | N/A |
| SRM-based*** | A | 97.73±0.14 | C | 10 / 0 | 436±17 |
| SRM-based*** | A | **98.30±0.06** | C | 60 / 10 | **6536±120** |
| ANTLR | A&T | 97.73±0.09 | C | 10 / 0 | 415±14 |
| ANTLR | A&T | **98.05±0.10** | C | 60 / 10 | 6638±130 |
| Timing*** | T | 94.10±0.51 | L | - | 2166±294 |
| ANTLR | A&T | 96.58±0.25 | L | - | **111±8** |

* A (activation-based), T (timing-based), and S (scalar-mediated, refer to Section 5), ** C (count loss) and L (latency loss), *** Our implementation of existing approaches

[Meta-Review · NeurIPS 2020]

This paper induced divergent reviews. Three of the reviewers felt it was a decent contribution that provided a novel insight about how to combine two different types of learning rule, but one reviewer felt strongly that it really did not provide any major contribution and only engaged in a relatively trivial mixing of models. Given these reviews, it was hard to come to a decision, but 'accept' seemed appropriate given that 3 out of 4 reviewers were fairly positive.